# Evolution and Dynamic Transcriptome of Key Genes of Photoperiodic Flowering Pathway in Water Spinach (*Ipomoea aquatica*)

**DOI:** 10.3390/ijms25031420

**Published:** 2024-01-24

**Authors:** Xin Wang, Yuanyuan Hao, Muhammad Ahsan Altaf, Huangying Shu, Shanhan Cheng, Zhiwei Wang, Guopeng Zhu

**Affiliations:** 1Key Laboratory for Quality Regulation of Tropical Horticultural Crops of Hainan Province, School of Breeding and Multiplication (Sanya Institute of Breeding and Multiplication), Hainan University, Sanya 572025, China; 181684@hainanu.edu.cn (X.W.); 19071001110001@hainanu.edu.cn (Y.H.); ahsanaltaf@hainanu.edu.cn (M.A.A.); shuhuangying@hainanu.edu.cn (H.S.); 990865@hainanu.edu.cn (S.C.); 2Key Laboratory for Quality Regulation of Tropical Horticultural Crops of Hainan Province, School of Tropical Agriculture and Forestry, Hainan University, Haikou 570228, China; 3Hainan Yazhou Bay Seed Laboratory, Sanya 572025, China

**Keywords:** water spinach, photoperiod, flowering, circadian rhythm, short day, long day

## Abstract

The photoperiod is a major environmental factor in flowering control. Water spinach flowering under the inductive short-day condition decreases the yield of vegetative tissues and the eating quality. To obtain an insight into the molecular mechanism of the photoperiod-dependent regulation of the flowering time in water spinach, we performed transcriptome sequencing on water spinach under long- and short-day conditions with eight time points. Our results indicated that there were 6615 circadian-rhythm-related genes under the long-day condition and 8691 under the short-day condition. The three key circadian-rhythm genes, *IaCCA1*, *IaLHY*, and *IaTOC1*, still maintained single copies and similar *IaCCA1*, *IaLHY*, and *IaTOC1* feedback expression patterns, indicating the conservation of reverse feedback. In the photoperiod pathway, highly conserved *GI* genes were amplified into two copies (*IaGI1* and *IaGI2*) in water spinach. The significant difference in the expression of the two genes indicates functional diversity. Although the photoperiod core gene *FT* was duplicated to three copies in water spinach, only *IaFT1* was highly expressed and strongly responsive to the photoperiod and circadian rhythms, and the almost complete inhibition of *IaFT1* in water spinach may be the reason why water spinach does not bloom, no matter how long it lasts under the long-day condition. Differing from other species (*I. nil*, *I. triloba, I. trifida*) of the *Ipomoea* genus that have three *CO* members, water spinach lacks one of them, and the other two *CO* genes (*IaCO1* and *IaCO2*) encode only one CCT domain. In addition, through weighted correlation network analysis (WGCNA), some transcription factors closely related to the photoperiod pathway were obtained. This work provides valuable data for further in-depth analyses of the molecular regulation of the flowering time in water spinach and the *Ipomoea* genus.

## 1. Introduction

*Ipomoea aquatica* Forsk is an annual or perennial herbaceous plant of the *Ipomoea* genus that is both aquatic and terrestrial. The stem is hollow and green or purple, and it has easy-to-grow adventitious roots at every node with high temperatures and moisture resistance [1]. Water spinach is widely cultivated in China and Southeast Asia, and it has high nutritional and medicinal value [2,3]. 

Flowering is an imperative process in the transformation from plant vegetative to reproductive growth, with complex regulatory mechanisms and pathways [4]. The flowering time is not only determined by genetic factors but is also initiated by environmental factors [5]. In recent years, with the continuous development of sequencing technology and genetic engineering, flowering-associated genes have been isolated and identified in a variety of plants, and the associated regulatory pathways of flowering have been clarified [6]. In the model plant *A. thaliana*, the flowering time is jointly controlled by the photoperiod pathway, vernalization pathway, autonomous pathway, and gibberellin pathway. The photoperiod pathway is the most thoroughly studied [7]. *FT* and *SOC1*, as key node genes, play a role in these pathways [8,9]. Water spinach naturally flowers in an inductive short day, which decreases the production of vegetative tissues and the food quality [2]. At present, the photoperiod-mediated regulation of water spinach flowering has not been clarified.

The molecular mechanism of the photoperiod regulation of plant flowering was discovered through the study of the model plant *A. thaliana* [5]. A photoperiod refers to the alternation of light and dark periods in the day–night cycle, which is an important environmental factor for plant flowering, and plants are separated into short-day plants, long-day plants, and day-neutral plants [10,11]. The photoreceptors in the plant leaves receive external light signals and convert the light signals into biological signals for the endogenous circadian clock [12]. The clock senses circadian changes to form a circadian rhythm and then initiates the expression of the *CO* gene, which controls downstream genes, such as *FT* and SOC1, to regulate plant flowering [13].

Photoreceptors can receive and recognize light signals of different wavelengths, and they are primarily distributed into three types: phytochromes, cryptochromes, and phototropins [14]. Phytochromes, including PHYA, PHYB, PHYC, PHYD, and PHYE, mainly absorb red light and far-red light in *A. thaliana* [12]. Cryptochromes (CRY1, CRY2, and CRY3) absorb blue and ultraviolet light [12,14]. The circadian clock plays a crucial role in the adjustment of the photoperiod [15]. Many circadian-clock components have transcriptional-regulation activity [16]. They are expressed sequentially over a period of about 24 hours, forming multiple feedback regulatory loops and regulating the expression of downstream genes to control plant growth, basal metabolism, and hormone metabolism [17,18,19].

In plants, the oscillatory mechanism involves a typical two-component response regulator named TOC1, or APRR1, and two related MYB transcription factors: LHY and CCA [20,21]. *TOC1* (*TIMING OF CAB EXPRESSION*), *LHY* (*LATE ELONGATED HYPOCOTYL*), *ELF4* (*EARLY FLOWERING 4*), *CCA1* (*CIRCADIAN CLOCK ASSOCIATED*), and *LUX* (*LUX ARRHYTHMO*) are crucial clock-associated genes, and the transcription and translation levels of the different genes constantly change, forming regular oscillations [9,22,23]. *LHY* and *CCA1* repress the expression of *TOC1* by binding to the *TOC1* promoter region. At the same time, TOC1 is a positive regulator of the expression of *LHY* and *CCA1* [24]. The inhibition of *TOC1* expression decreases the expression of *LHY* and *CCA1*. *LHY* and *CCA1* repress the transcription of the *TOC1* gene throughout the day. The expression of the TOC1 protein at night up-regulates the transcription of *LHY* and *CCA1*, resulting in peak expression in the morning [25,26,27]. There are both positive and negative feedback loops between *TOC1* and *LHY*/CCA1, which may be the key to the circadian rhythm. These circadian-clock-related genes are located upstream of the photoperiod regulation pathway, and their continuous changes lead to changes in the expression of the downstream genes, which affect the flowering times of plants [19,24]. ELF3 and GI enable oscillators to synchronize endogenous cellular mechanisms with external environmental signals. The absence of the *ELF3* and *GI* genes in plants affects clock-mediated photoperiod-responsive growth and development. This shows that the ELF3 and GI genes work together to transmit photoperiod sensing to the central oscillator [28].

The *CO* gene is a crucial part of the photoperiod pathway and the biological-rhythm pathway. It can convert light signals into flower signals and affect the flowering times of plants [29]. In *A. thaliana*, the CO protein is a nuclear zinc-finger protein containing B-box and CCT domains, which can bind to the promoter of *FT* and activate flowering [29,30], and multiple *CO* genes have been identified in plants [31,32,33,34]. The expression of *CO* is regulated by the circadian clock and light signal pathways. The transcription of the *CO* gene is primarily controlled by CDFs and the FKF1-GI complex [35]. CDF transcription factors, including CDF1, CDF3, CDF4, and CDF5, act in concert with the CO promoter and repress transcription. CDF1 has been demonstrated to directly bind to the *CO* regulatory regions and act as a repressor of CO transcription [36]. Further studies have found that CDF can up-regulate the expression of the transcription repressor TPL, thus repressing the expression of the *CO* gene [35,37]. FKF1 is a kind of E3 ubiquitination ligase. FKF1 forms a complex with GI and activates the transcription of CO by degrading CDF, a repressor of *CO* transcription [35,36,37,38,39,40]. The bHLH transcriptional activators FBH1, FBH2, FBH3, and FBH4 recognize and bind to the E-box cis-element in the CO promoter and then activate the transcription of *CO*. Overexpression of the FBH protein can up-regulate *CO* levels and cause early flowering [41].

*CO* is regulated not only at the transcription level but also at the level of its protein stability and accumulation. The E3 ubiquitin ligase COP1 (CONSTITUTIVELY PHPYOMORPOGENIC 1) inhibits flowering by promoting the hydrolysis of the CO protein at night [42,43]. In plants, the E3 ligase activity of COP1 depends on its interaction with SUPPRESSOR OF PHYA-105 (SPA) proteins. The four-member SPA protein family of *A. thaliana* acts in concert with COP1 to inhibit photomorphogenesis at night [10,44,45]. The CDF1 protein binds to the FT promoter to repress FT transcription, while FKF1 can remove CDF1 from the FT promoter region to initiate *FT* transcription [46]. *CO* stimulates FT expression by binding to CO-responsive elements (COREs) and CCAAT-box elements in its promoter [32,47].

*FT* is a member of the phosphatidylethanolamine-binding protein (PEBP) family, and the other two homologs, *TSF* and *MOTHER OF FT* (*MFT*), have a redundant function as flowering inducers [48,49]. The *FT* gene is a key factor in the photoperiod pathway, and its expression is closely associated with the flowering time. As *FT* activators, *CIB*s interact directly with the *FT* promoter, initiating its expression. CRY2 responds to blue light, forms a protein complex with CIB1 and CO, and promotes flowering [50,51]. GI can bind to three FT repressors (SHORT VEGETATIVE PHASE (SVP), TEMPRANILLO (TEM)1, and TEM2) to regulate *FT* expression in a CO-independent manner [52]. *FLC* and *SHORT VEGETATIVE PHASE* (*SVP*) directly suppress *FT* and *SOC1* expression [53]. The FT protein is first produced in the companion cells in the phloem and then transmitted to the shoot apical meristem, where it forms a complex with the bZIP transcription factor FD (FLOWERING LOCUS D) [54], activating the expression of the floral meristem identity gene *APETALA1* (*AP1*); meanwhile, *LEAFY* (*LFY*) is first up-regulated in leaf primordia and the SAM. *FRUITFUL* (*FUL*) is later activated and eventually initiates the flowering process [11,33,55].

In this study, we used an RNA-seq sequencing platform to find circadian-rhythm-related genes, determine the expression patterns of flowering-associated genes, and discover the regulatory network of flowering genes through weighted correlation network analysis (WGCNA). The objectives of this study were to (1) identify homologous genes associated with flowering in water spinach and (2) preliminarily identify the photoperiod-mediated molecular regulation mechanism of flowering time in water spinach.

## 2. Results

### 2.1. RNA-Seq Data Statistics

We performed RNA sequencing at different time points from different photoperiods and constructed 48 libraries. A total of 215.10 million raw reads were generated through RNA-seq. Paired-end sequences with low-quality reads were filtered out. Finally, 205.92 million clean reads were obtained. The error rate was less than 0.03%, on average; the GC content was 45.57%, and Q30 (the percentage of bases with Phred values greater than 30 bases in total) was 94.06%. Interestingly, 95.79% of reads were mapped to the *Ipomoea aquatica* reference genome, and about 91.64% were mapped to unique positions. Among these reads, more than 82.04% were distributed in exon regions, 6.85–10.29% were distributed in intronic regions, and 3.52–7.91% were distributed in intergenic areas.

### 2.2. Analysis of Photoperiod-Related Genes and Circadian-Rhythm-Related Genes

In our experiment, the data thresholds were a *p*-value < 0.05 and |log2FoldChange| > 0.5. A total of 6821 differentially expressed genes were separated and named photoperiod-related genes, of which 2698 were up-regulated and 4123 were down-regulated. GO enrichment showed that in the CC fraction, the genes were mainly enriched in terms related to the photosystem. Through KEGG pathway annotation, photosynthesis, cutin, suberine, wax biosynthesis, and ribosome biogenesis in eukaryotes were significantly enriched (Figure 1).

We summarized the differentially expressed genes in all comparisons at different time points in the long day and short day and named them circadian-rhythm-related genes. The results indicated that there were 9951 circadian-rhythm-related genes in total: 6615 circadian-rhythm-related genes under the long-day condition and 8691 circadian-rhythm-related genes under the short-day condition, with 5355 rhythm-related genes in common between them (Figure 2A). The KEGG enrichment results indicated that the 5355 genes were primarily enriched in the circadian-rhythm–plant pathway (Figure 2B). For circadian-rhythm-related genes, we deleted genes with FPKM < 10 in all samples based on their expression levels; we called a gene whose expression level is higher in light than in the dark a light-induced gene and a gene whose expression level in the dark is higher than that in light a dark-induced gene. The results indicated that under long-day conditions, there were 151 light-induced genes and 185 dark-induced genes. Under short-day conditions, there were 483 light-induced genes and 277 dark-induced genes (Figure 2C). Further, we investigated the relationship between photoperiod-related genes and circadian-rhythm-related genes. The Venn diagram showed that 3687 genes were associated with both the photoperiod and circadian rhythm, 3134 genes were only related to the photoperiod, and 6264 genes were only associated with circadian rhythms (Figure 2D).

### 2.3. Molecular Network of Core-Clock Oscillator

The feedback regulation between *LHY/CCA1* and *TOC1* genes was the key to the circadian rhythm. Through phylogenetic analysis, there was only one copy of *CCA1*, *LHY*, and *TOC1* in water spinach, namely, *iaq.scaf_129.1643_iaq.scaf_129.1644* (*IaCCA1*), *iaq.scaf_51.23* (*IaLHY*), and *iaq.scaf_884.166* (*IaTOC1*). The transcriptome results specified that *IaCCA1* and *IaLHY* gene expression patterns were similar and exhibited the same oscillation pattern under different photoperiods, with the strongest suppression in the dark for four hours under the short-day condition. The *IaTOC1* transcription level was opposite to those of *IaCCA1* and *IaLHY*, with the expression peak occurring after four hours of darkness under the short-day condition (Figure 3). These research results confirmed the negative regulatory mechanism of *LHY/CCA1* and *TOC1.*

### 2.4. Analysis of Photoperiod-Mediated Flowering-Related Genes

The photoperiod-mediated plant flowering time regulation pathway has been mainly elucidated in *A. thaliana* (Figure 4). The photoreceptors regulate the expression of *CO* via *GI*, a gene involved in the circadian rhythm clock, and *CO* then activates the downstream gene *FT*. The *GI-CO-FT* model is the major photoperiod pathway in *A. thaliana.*

The transcription of *GI* is controlled by the circadian clock, and the phylogenetic tree reveals that *GI* has two copies in water spinach and that the transcript level of *iaq.scaf_28.25* (*IaGI1*) was higher than that of *iaq.scaf_115.70* (*IaGI2*), signifying the main role of *iaq.scaf_28.25* as the ortholog of GI. The expression level of *iaq.scaf_28.25* displayed a clear circadian rhythm oscillation, and the transcript level increased just after light and decreased after dark (Figure 5).

Phylogenetic analysis showed two CO subgroup members, iaq.scaf_9.675 (IaCO1) and iaq.scaf_940.190 (IaCO2), in water spinach and three in *I. nil* (XM_019330139.1, XM_019320846.1, and XR_002055520) (Figure 6A). The domain prediction revealed only one B-box in iaq.scaf_940.190 and iaq.scaf_9.675. In contrast to XM_019330139.1, encoding a protein with two B-boxes and one CCT domain, the *iaq.scaf_9.675* gene encodes a truncated protein containing only one B-box due to an eight-base deletion of *aq.scaf_9.675* (Figure 6B). The homozygous *iaq.scaf_9.675* locus was confirmed by Illumina reads. Genome BLAST and orthologous region alignment showed that the XR_002055520 ortholog was deleted from the water spinach genome (Figure 6C).

In order to identify the gene that involves *CO*, we explored two CO subgroup numbers in water spinach. In the short-day condition, the transcript level of *iaq.scaf_9.675* progressively decreased during the day, and when darkness came, the expression progressively increased, with the expression peak in the early morning. Additionally, we found that the *iaq.scaf_940.190* gene exhibited low expression (Figure 6D). Hence, it can be speculated that the *iaq.scaf_9.675* gene plays a role as a *CO* homolog in water spinach. To further investigate the expression pattern of the *iaq.scaf_9.675* gene, we further added a sampling time point, 5 min before light (S_24h and L_24h); real-time quantitative PCR showed that the transcript level of *iaq.scaf_9.675* accumulated continuously at night, and the transcript level decreased rapidly when exposed to light, signifying that the transcript level of *CO* was very sensitive to light (Figure 6E). 

The *A. thaliana FT/FTL1* gene family contains 6 genes, and the family is expanded to 12 members with 3 *FT* subgroup members in water spinach (*iaq.scaf_129.59* (*IaFT1*), *iaq.scaf_323.202*, and *iaq.scaf_323.203*). During the long day, almost no transcripts were detected for any members in the leaves. During the short day, only the *FT* subgroup members were strongly up-regulated (Figure 7). The expression of the *iaq.scaf_129.59* gene was significantly higher than the others, and its expression gradually decreased with the increase in daylight time and then gradually increased after darkness, reaching the peak in the early morning. These results suggest that *iaq.scaf_129.59* is an ortholog of *FT* (Figure 7C). The loss of expression of *iaq.scaf_129.59* could be the cause of the complete inhibition of the flowering ability of water spinach under the long day length.

Moreover, we investigated the genes associated with the flowering mechanism in water spinach through phylogenetic and comparative transcriptome analyses, and the outcomes indicate that there are two copies of *AGL24*, *SVP*, *SOC1*, *FUL*, and *AP1* in *Ipomoea* species, suggesting that these genes are linked to a common ancestor. *COP1*, *LFY*, and *FPK1* have only one copy in *Ipomoea species* (Appendix A). 

*iaq.scaf_1261.31* and *iaq.scaf_913.48* are *FKFI*-homologous genes; *iaq.scaf_19.19* accumulated under both long-day and short-day conditions, and the transcription level was consistent. The expression progressively increased with the daylight hours, reaching a peak in the evening, and then the transcript level continuously declined in the dark, signifying a similar expression pattern to that of *iaq.scaf_28.25* (*IaGI*). As a negative regulator of flowering, the transcript level of *iaq.scaf_56.407* (*IaCOP1*) was opposite to those of the *IaGI* and *IaFPK1* genes: the transcript level decreased continuously during the day, and the expression level gradually accumulated after dark (Appendix A). 

As homologous genes of *AGL24* in water spinach, the expression patterns were similar, and they were both induced by short-day conditions. The gene transcript level under the short-day condition was significantly lower than that under the long-day condition. There are two *SOC1* homologs (*iaq.scaf_56.882* and *iaq.scaf_130.151*) in water spinach, and the transcript levels of the two genes exhibited regular oscillations. With the increase in light time, the transcript levels gradually decreased. After dark, the transcript levels gradually increased again, reaching a peak in the early morning, but only *iaq.scaf_130.151.1* was induced by a short day. As SVP homologs, the transcript level of *iaq.scaf_940.636.1* was higher than that of *iaq.scaf_9.539*. In addition, the expression of the *iaq.scaf_940.636.1* gene was slightly higher under long-day conditions. Presumably, this gene had a repressive effect on the transcript levels of *SOC1* and *FT*. As *FUL* homologs, *iaq.scaf_107.214* and *iaq.scaf_884.85* had high transcript levels and were strongly induced by short days. Among them, the transcript level of *iaq.scaf_884.85* was higher than that of *iaq.scaf_107.214*, suggesting that *iaq.scaf_884.85* plays a significant part in the flowering pathway. Together, the transcription levels indicated that *LFY*- and *AP1*-homologous genes in water spinach were not expressed under different light conditions (Appendix A).

### 2.5. Validation of RNA-Seq Data by qRT-PCR

We selected eight flowering-related genes for qRT-PCR analysis to confirm the accuracy of the RNA-seq results. The results indicated that the relative expression levels of the samples determined by qRT-PCR were generally consistent with the FPKM values of the samples analyzed by RNA-seq (Appendix A), thus representing the stability and consistency of the RNA-seq data.

### 2.6. WGCNA Co-Expression Network Analysis

To analyze the correlations between the expression levels of genes, we divided the genes into different modules, where genes in the same module may have similar biological functions. We performed a weighted gene co-expression network analysis (WGCNA), and the co-expressed genes were clustered into 37 major tree branches, each of which represents a module (labeled with different colors) (Figure 8A).

The analysis of the module–trait relationship revealed that the yellow module was negatively associated with dark time (r = −0.81, *p* = 1 × 10^−4^), while the red module was positively associated with dark time (r = 0.95, *p* = 2 × 10^−8^). The pink module was positively correlated with the short-day treatment (r = 0.93, *p* = 2 × 10^−7^). The FT candidate gene *iaq.scaf_129.59* was in the dark-magenta module, and the module had a high positive correlation with S_0h and S_3h (Figure 8B,C).

In addition, heatmaps were drawn to demonstrate the gene expression pattern within each module. The gene expression patterns were opposite between the yellow module and the red module. The genes in the yellow module were highly expressed in light, whereas the genes in the red module had lower expression levels during the day. The length of sunlight was associated with genes in the pink module, and most of the genes were highly expressed under the long-day condition. The dark-magenta-module genes only showed high expression at S_0h and S_3h, which shows that the module genes were stimulated by light, and the expression level increased sharply (Figure 9).

In a module, genes with high connectivity (K for the connectivity of genes within modules) were distinguished as hub genes. In the four modules, the top 20 genes with high K were selected as the hub genes. Additionally, we observed that some transcription factors were among these genes. Six genes belonged to the cytochrome P450 family in the dark-magenta module (Table 1).

To further explore the biological functions of the modules, we completed a KEGG analysis of the genes in the four modules. KEGG analysis revealed that the genes in the yellow module were enriched in lysine degradation, circadian-rhythm–plant, carbon metabolism, and RNA degradation pathways; the iaq.scaf_28.25 (GI-like), iaq.scaf_82.76 (LFY-like), and iaq.scaf_56.882 (SOC1-like) genes were present in this module. In the red module, the genes were significantly enriched in alanine, aspartate, and glutamate metabolism; tyrosine metabolism; and circadian-rhythm–plant pathways. Together, the yellow- and red-module genes were significantly enriched in the circadian-rhythm–plant pathway, which indicated that this pathway was significantly related to the length of light. In the pink module, the genes were greatly enriched in oxidative phosphorylation; ribosome; glycine, serine, and threonine metabolism; tyrosine metabolism; and isoquinoline alkaloid biosynthesis pathways. In the dark-magenta module, the genes were significantly enriched in sulfur metabolism, alpha-linolenic acid metabolism, base excision repair, and galactose metabolism pathways (Appendix A).

There are some flowering-associated genes in these four modules: iaq.scaf_28.25 (GI-like), iaq.scaf_115.70 (GI-like), iaq.scaf_82.76 (LFY-like), and iaq.scaf_56.882 (SOC1-like) in the yellow module; iaq.scaf_56.407 (COP1-like), iaq.scaf_9.539 (SVP-like), iaq.scaf_51.23 (LHY-like), iaq.scaf_323.203 (FT-like), and iaq.scaf_940.190 (CO-like) in the red module; iaq.scaf_913.48 (AGL24-like), iaq.scaf_857.108 (AP1-like), iaq.scaf_107.214 (FUL-like), and iaq.scaf_884.85 (FUL-like) in the pink module; and the iaq.scaf_129.59 (FT-like) gene in the dark-magenta module. To further determine the relationships between flowering-associated genes in the four modules, correlation networks were constructed.

In the yellow module, the co-expression network shows that iaq.scaf_28.25 (GI-like), iaq.scaf_115.70 (GI-like), and iaq.scaf_82.76 (LFY-like) had a strong interaction relationship, with seven genes interacting with these three genes at the same time. As the hub gene, iaq.scaf_9.878 interacted with the four flowering-related genes, and the annotation information showed that this gene belongs to the G6PD transcription factor family (Figure 10A).

In the red module, there were no correlations between iaq.scaf_9.539 (SVP-like) and other flowering-associated genes, suggesting that iaq.scaf_9.539 (SVP-like) plays an independent role in the flowering regulation pathway. iaq.scaf_51.23 (LHY-like), iaq.scaf_940.190 (CO-like), and iaq.scaf_56.407 (COP1-like) had a high correlation; it is speculated that the three genes play a synergistic role in the regulation of flowering. iaq.scaf_111.226 was interrelated with the four flowering-related genes (Figure 10B).

In the pink module, the analysis indicated that a strong interaction exists between iaq.scaf_107.214 (FUL-like) and iaq.scaf_884.85 (FUL-like). Furthermore, iaq.scaf_10.357 and iaq.scaf_50.99 each interact with three flowering-related genes, and it is speculated that they may be involved in flowering regulation (Figure 10C).

In the dark-magenta module, there were a total of 89 genes interacting with iaq.scaf_129.59 (IaFT1), including 41 transcription factors (symbolized by triangles), and 6 genes were correspondingly enriched in diverse KEGG pathways (Figure 10D).

## 3. Discussion

Flowering is involved in the transition from vegetative growth to reproductive growth and is similarly a crucial trait in agricultural production. The response of *A. thaliana* flowering to the photoperiod has been studied widely and in great depth, and many genes controlling photoperiodic flowering have been characterized [52,56,57]. In our work, we identified homologs of several critical genes in the photoperiod pathway regulating the flowering time in water spinach. 

There is positive and negative feedback regulation between LHY/CCA1 and TOC1 [25,58]. CCA/LHY stimulates PPR7 and PPR9 transcription throughout the day, whereas PPR7 and PPR9 bind to the CC1 promoter region and constrain CCA1 expression [59]. CCA1 and LHY limit the transcription of clock-associated genes; CCA1 and LHY protein levels decline at the end of the day, and their inhibitory action no longer takes place on their target genes, leading to the accumulation of the evening-clock-gene transcripts [15,59]. The evening expression of TOC1 represses GI, which, in turn, triggers TOC1 and forms the evening loop [28,59].

The transcriptome data revealed that the transcript level of iaq.scaf_884.166 (TOC1-like) began to rise after light, whereas the expression levels of iaq.scaf_51.23 (LHY-like) and iaq.scaf_129.1643_iaq.scaf_129.1644 (CCA1-like) gradually decreased. The transcription levels of Ziaq.scaf_51.23 (LHY-like) and iaq.scaf_129.1643_iaq.scaf_129.1644 (CCA1-like) were accompanied by a decrease in the iaq.scaf_884.166 (TOC1-like) expression level in the dark. These outcomes validated the negative regulatory effect of LHY/CCA1 on TOC1 [28]. In addition, *LHY* and *CCA1* also regulate the transcription of *GI* and *FKF1* genes [60]. In our work, there was a negative correlation between *iaq.scaf_51.23* (*LHY-like*), *iaq.scaf_129.1643_iaq.scaf_129.1644* (*CCA1-like*), and *iaq.scaf_28.25* (*GI-like*) based on the transcription levels, which established that *LHY/CCA1* inhibited the transcription of *GI* [60]. *LHY/CCA1* accelerated flowering by promoting FT expression and reducing the abundance of SVP [16]. The *TOP1* gene was highly expressed at night and suppressed the transcriptional levels of *LUX*, *ELF4*, and *GI* [59], and the transcriptome data confirmed this result. It has been shown that there is a redundant function between the *LHY* and *CCA1* genes [55]. In addition, the lack of *CCA1* also affects the regulation of gene expression by phytochromes, suggesting that *CCA1* has an independent function [61]. Recent research revealed that the expression of CCA1/LHY was regulated by Brassinosteroids (BRs) [62]. DET1, as an important core transcriptional repression factor, regulates the clock process by interacting with CCA1/LHY target genes [58]. *CCA1* and *LHY* also have the function of coordinating Fe homeostasis in *A. thaliana* by directly regulating Fe metabolism and the transcription of transport genes [63]. 

The expression of *GI* is under the control of the circadian clock and peaks at the end of the day, leading to the optimal formation of the GI–FKF1 complex, which degrades CDF proteins and, in turn, leads to the induction of *CO* [4,39]. In our study, the expression of *GI* and *FKF1* peaked at the end of the day, and *CO* accumulated at night, which is consistent with the previous results. The transcriptome data in our studies showed that *CO* expression was inhibited after light, probably due to *CCA1* and *LHY* positively regulating the expression of *CDF1* at dawn and the inhibition of *CO* by *CDFs* and *FLOWERING BHLH* (*FBH*) [41,64]. Phylogenetic analysis revealed that a member of the CO subgroup was lost in water spinach; whether the gene plays a role in regulating flowering remains to be determined. Day length must reach a threshold in order to promote the stabilization of the CO protein. The CO protein is not degraded and accumulates during the night in *A. thaliana* [4,64]. The transcriptome data showed that the expression level of *iaq.scaf_9.675*(*IaCO1*) was higher during short days and accumulated only on short-day nights, which had a positive effect on plant flowering. Moreover, the CCT domain was deleted in iaq.scaf_9.675 (CO-like). The CCT domain has significant functions in transcriptional regulation and nuclear protein transport. CO undergoes alternative splicing, producing the full-size COα, which is equivalent to the well-known CO protein, and the C-terminally truncated COβ lacking the CCT domain. COβ inhibits COα function by forming heterodimers in photoperiodic flowering [5], though it confirms that the function of CO-like genes lacking the CCT domain could play a role in water spinach. *A. thaliana* CO is a member of the BBX family with 17 proteins. BBX2 and BBX3 are highly similar to CO and produce only slight effects on photoperiodic flowering. CO contains two B-boxes and one CCT domain required for flowering regulation in *A. thaliana* [32].

Here, three members from the FT subgroup were identified in water spinach; according to the gene expression level, we infer that *iaq.scaf_129.59* performs the role of the *FT* gene in water spinach. In the meantime, in our study, the abundance of *AP1* and *LFY* was very low, and FUL was highly expressed in the short-day condition; these would be the critical genes for flowering in water spinach. SVP plays a crucial function in directly controlling SOC1 transcription, while *FT* expression in the leaf is slightly modulated by *SVP* [53]. In our work, the transcriptome data of water spinach treated at different times were examined by using WGCNA to construct the co-expression network of flowering-associated genes. The co-expressed genes were separated into 37 modules via hierarchical average linkage cluster analysis.

Here, we found that the red module and yellow module have positive and negative correlations with the darkness duration, respectively. These genes are mainly enriched in the circadian-rhythm–plant pathway, according to KEGG analysis; these genes are also enriched in the carbon metabolism and amino acid metabolism pathways, indicating that the circadian clocks are strongly associated with plant growth [13,65].

A previous study reported that GI mediates the circadian clock and the three floral integration genes *FT*, *SOC1*, and *LFY* to control photoperiodic flowering [66]. In our study, there was a strong correlation between the *GI*, *SOC1*, and *LFY* genes, which further elucidates that GI has a regulatory impact on SOC1 and LFY. We observed that the transcription level of *GI* reaches its peak at about 8 hours after dawn, which is in line with a previous study [67]. The gene co-expression network showed that the cytochrome P450 transcription factor family is closely associated with iaq.scaf_129.59 (FT-LIKE1). It is shown that the P450 gene family is involved in the regulation of FT-mediated flowering pathways, which needs to be analyzed in future studies. 

## 4. Materials and Methods

### 4.1. Plant Materials and Sampling

In this study, the water spinach germplasm HNUWS004 (planted at Hainan University, Haikou, China) was used as material and propagated using the hydroponic method. The plants were treated with short-day and long-day conditions in separate growth chamber incubators. During the whole experiment, the long-day treatment condition was 16 h light/8 h dark (28 °C), and the short-day treatment condition was 8 h light/16 h dark (28 °C). We separated 24 h into 8 different time points and collected samples every 3 h. Under the long-day condition, the time points were as follows: light 0 h (L_0h), light 3 h (L_3h), light 6 h (L_6h), light 9 h (L_9h), light 12 h (L_12h), light 15 h (L_15h), darkness 2 h (L_18 h), darkness 5 h (L_21h). Under the short-day condition, the time points were the following: light 0 h (S_0h), light 3 h (S_3h), light 6 h (S_6h), darkness 1 h (S_9h), darkness 4 h (S_12h), darkness 7 h (S_15h), darkness 10 h (S_18h), darkness 13 h (S_21h). After 3 weeks, as the plants reached a length of about 30 cm, fresh leaf samples (from the top) were collected from three plant replicates at 8 time points during the day and then stored in the refrigerator at −80 °C for subsequent RNA extraction.

### 4.2. RNA Quantification and Qualification

Total RNA was extracted from the leaves of water spinach, and the RNA concentration and integrity were evaluated using Bioanalyzer 2100 equipment (Agilent Technologies, Santa Clara, CA, USA). Using an Illumina NovaSeq platform, the libraries were sequenced, generating 150 bp paired-end reads. Low-quality reads, reads containing poly-N, and reads containing adapters were eliminated from the raw data to produce high-quality reads. Clean reads were aligned to the reference genome using Hisat2 (USA, CCB at JHU, v2.0.5). FPKM (expected number of fragments per kilobase of transcript sequence per million base pairs sequenced of each gene) was determined based on the length of the gene and the read count mapped to this gene [68].

### 4.3. Differential Expression Gene Analysis

Differential expression analysis between samples was finalized using the DESeq2 R package (1.20.0). The subsequent *p*-values were adjusted using Benjamini and Hochberg’s method. Genes with an adjusted *p*-value < 0.05 initiated by DESeq2 were assigned as differentially expressed and subjected to enrichment analysis [69].

### 4.4. GO and KEGG Enrichment Analysis

GO (Gene Ontology) is a comprehensive database for unfolding gene functions, which can be divided into three categories: biological process (BP), cellular component (CC), and molecular function (MF). GO and KEGG pathway enrichment analyses were performed using the cluster Profiler R package [70,71].

### 4.5. Weighted Gene Co-Expression Network Analysis (WGCNA)

WGCNA is a technique that can be used to find genes that unveil similar patterns of expression and potentially have functions in specific biological processes [72]. A gene co-expression network was built using the WGCNA package in R, with R^2^ > 0.8 as the criterion. A topological overlap matrix was calculated by comparing the connectivity similarities of each pair of probes among all genes. Furthermore, an eigen-gene network was assembled to signify the relationships among the modules. The networks were visualized by using Cytoscape (v.3.8.1) [73].

### 4.6. Sequence Alignment and Phylogenetic Tree Construction

The amino acid sequence of the *A. thaliana* flowering gene was downloaded from NCBI, and Bioedit software (V7.2.5) was used for the homologous gene alignment of flowering-related genes in water spinach, *I. triloba*, *I. trifida*, *I. nil*, *and C.australis*. The phylogenetic tree was constructed using the MEGA 11 method with 1000 bootstrap replicates [4].

### 4.7. Real-Time Quantitative PCR Analysis

Real-time quantitative PCR was used to test the reliability of the RNA-seq data. We selected 10 flowering-related genes for RT-qPCR validation. RNA was isolated from leaf samples from water spinach and transcribed into cDNA. GAPDH was used as an internal reference gene, and primers were designed based on the NCBI database. Three technical replicates were performed for each sample. Appendix A contains gene information for qRT-PCR analysis. The related quantitative data were calculated according to the 2^−∆∆CT^ method [73].
ΔCT(test) = CT(target, test) − CT(ref, test)
ΔCT(calibrator) = CT(target, calibrator) − CT(ref, calibrator)
ΔΔCT = ΔCT(test) − ΔCT(calibrator)

## 5. Conclusions

In sum, we performed a comprehensive transcriptome analysis of water spinach at different time points under short-day and long-day conditions, explored some flowering-associated genes, and analyzed their expression levels. Furthermore, the co-expression networks of flowering-associated key genes were generated using WGCNA. Overall, our results reveal the complex flowering regulation network of water spinach and lay the biological foundation for the regulation of water spinach flowering, which could be utilized in other horticulture plants. 

## Figures and Tables

**Figure 1 ijms-25-01420-f001:**
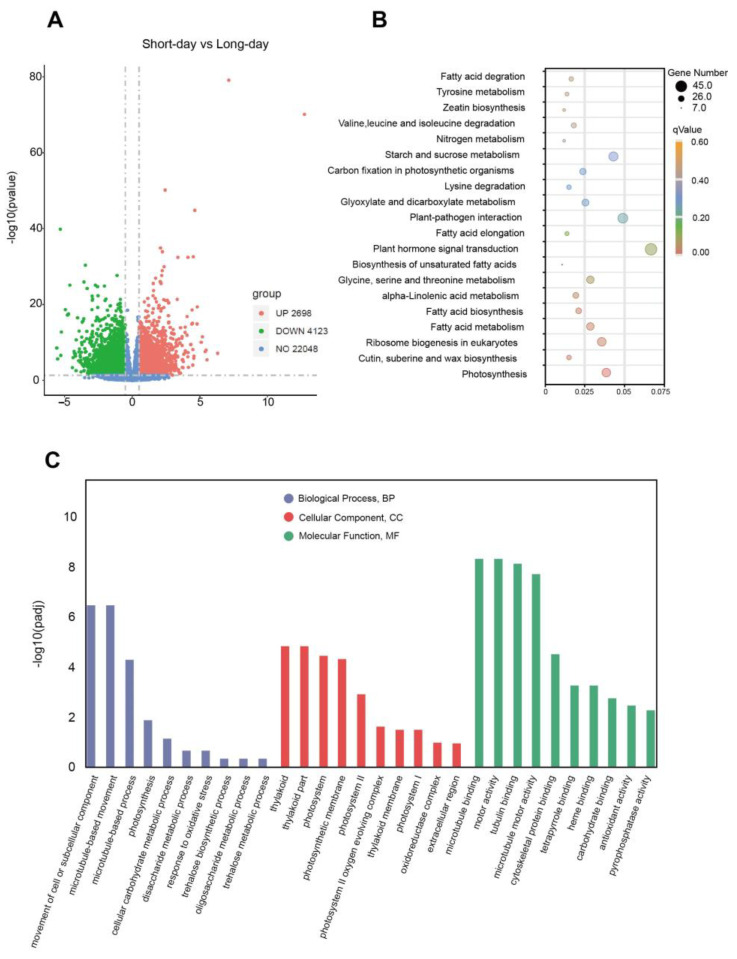
GO and KEGG enrichment analysis of differentially expressed genes in water spinach under short-day and long-day conditions: (**A**) volcano plot indicating the DEGs under short-day and long-day conditions; (**B**) KEGG pathway enrichment of photoperiod-related genes; (**C**) GO functional enrichment of photoperiod-related genes. Three main GO categories are summarized: BP, CC, and MF. BP, biological process; CC, cellular component; MF, molecular function. The number of DEGs is shown on the bar.

**Figure 2 ijms-25-01420-f002:**
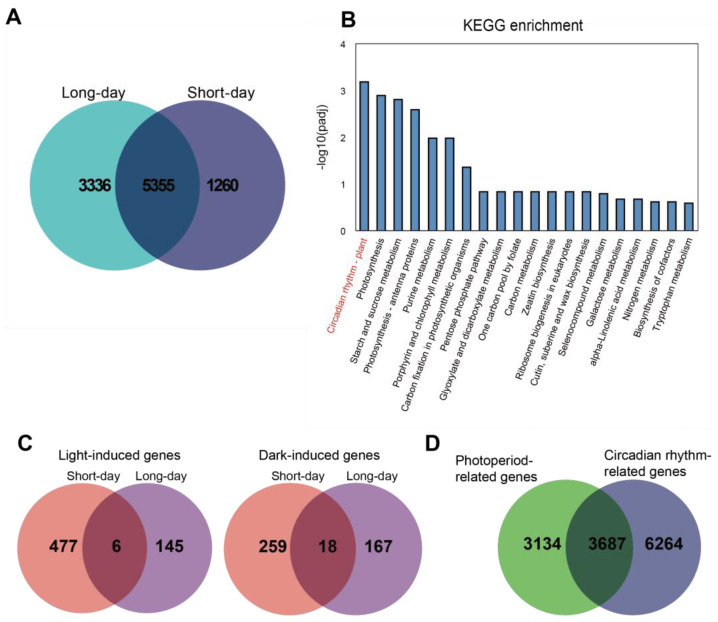
(**A**): Venn diagram representation of DEGs in long-day and short-day conditions; (**B**): KEGG enrichment analysis of 5355 circadian-rhythm-related genes; (**C**): Venn diagram representation of light-induced genes and dark-induced genes; (**D**): Venn diagram representation of photoperiod-related genes and circadian-rhythm-related genes.

**Figure 3 ijms-25-01420-f003:**
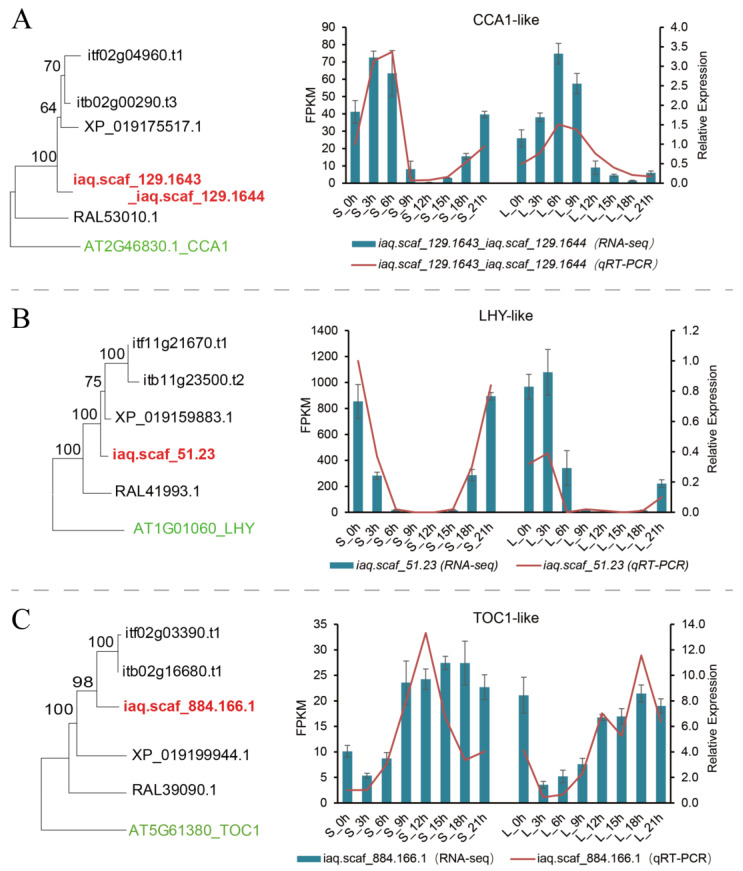
Phylogenetic analysis and transcript level of *CCA1* (**A**), *LHY* (**B**), and *TOC1* (**C**) in water spinach. S indicates short day; L indicates long day.

**Figure 4 ijms-25-01420-f004:**
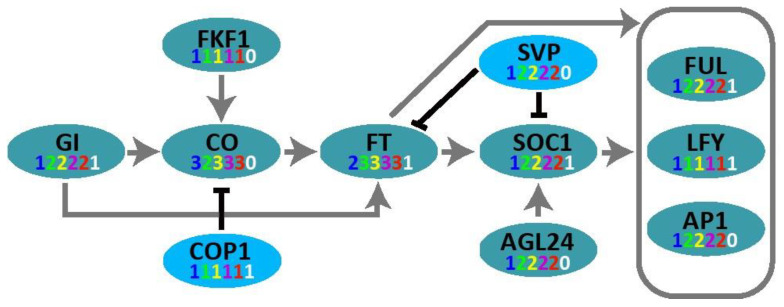
Simplified photoperiod pathway controlling flowering time. Numbers in blue, green, yellow, purple, red, and white represent copy numbers in *A. thaliana*, water spinach, *I. triloba*, *I. trifida*, *I. nil*, and *C. australis*, respectively. Arrows and T-ends represent the promotion and inhibition of gene expression, respectively.

**Figure 5 ijms-25-01420-f005:**
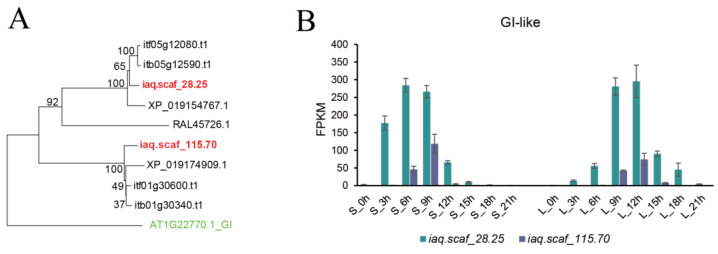
Phylogenetic analysis (**A**) and transcript level (**B**) of *GI* in water spinach. S indicates a short day; L indicates a long day. Results of transcriptome analysis during the different time courses are shown.

**Figure 6 ijms-25-01420-f006:**
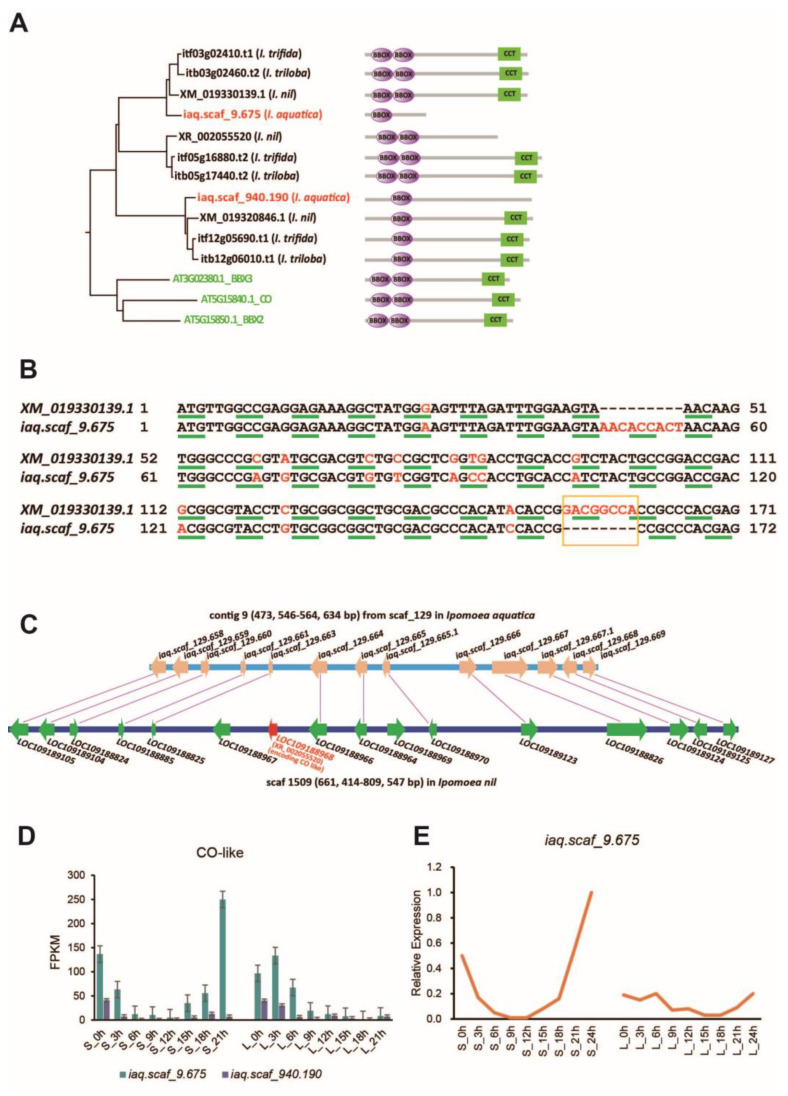
(**A**): Phylogenetic analysis and domain prediction of the CO subgroup members; (**B**): alignment of *iaq.scaf_9.675* (*IaCO1*) and its ortholog XM_019330139.1; (**C**): synteny analysis between water spinach orthologs and the *I. nil* genome containing *XR_002055520* encoding *CO-like*; (**D**): transcript level of *CO-like* genes in water spinach; (**E**): relative expression of *iaq.scaf_9.675*. S indicates short day; L indicates long day.

**Figure 7 ijms-25-01420-f007:**
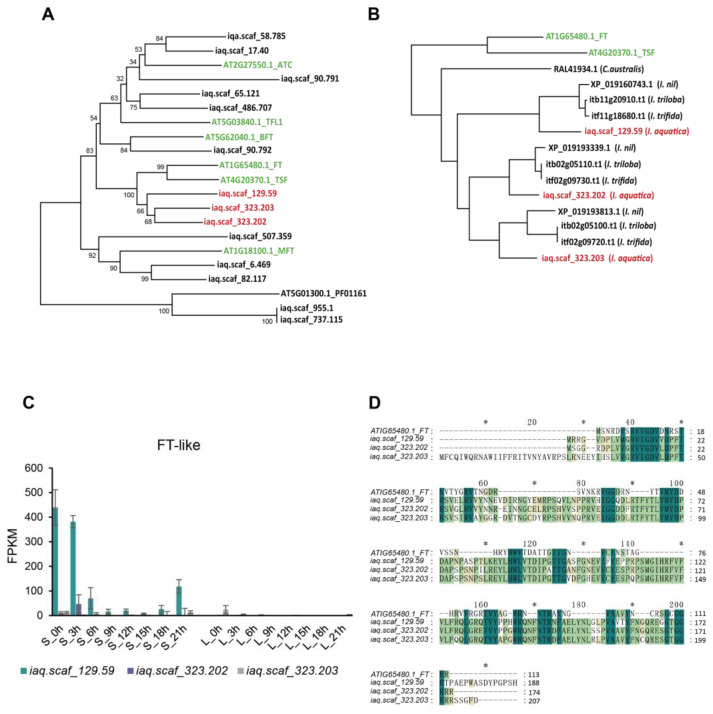
(**A**): Phylogenetic analysis of FT/FTL1 gene family; (**B**): phylogenetic analysis of the FT subgroup members; (**C**): transcript levels of *FT-like* genes in water spinach; (**D**): multiple sequence alignment based on amino acid sequences. S indicates short day; L indicates long day. * asterisk in the middle of the figure representing 10 amino acids, which are 10, 30, and 50 respectively.

**Figure 8 ijms-25-01420-f008:**
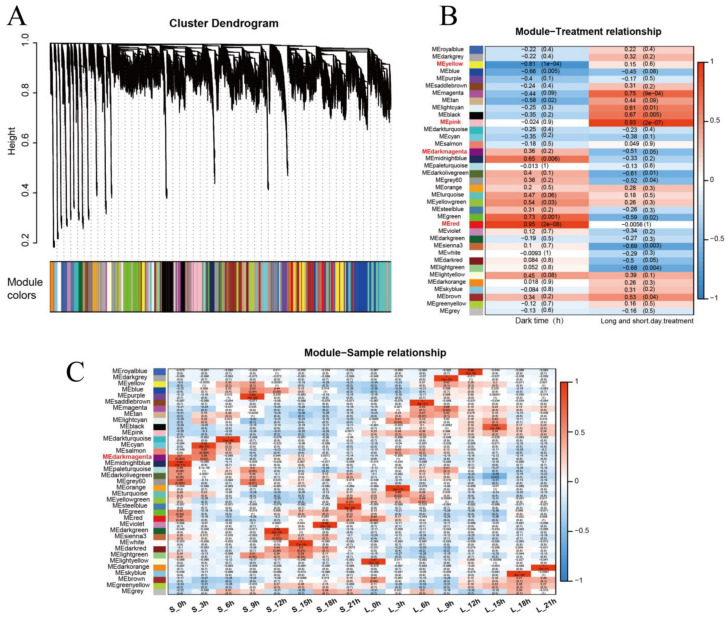
Gene co-expression network analysis by WGCNA (**A**): hierarchical cluster of 37 modules of co-expressed genes; (**B**): module–trait relationship heatmap for different traits and modules; (**C**): module–sample relationship heatmap. S indicates short day; L indicates long day.

**Figure 9 ijms-25-01420-f009:**
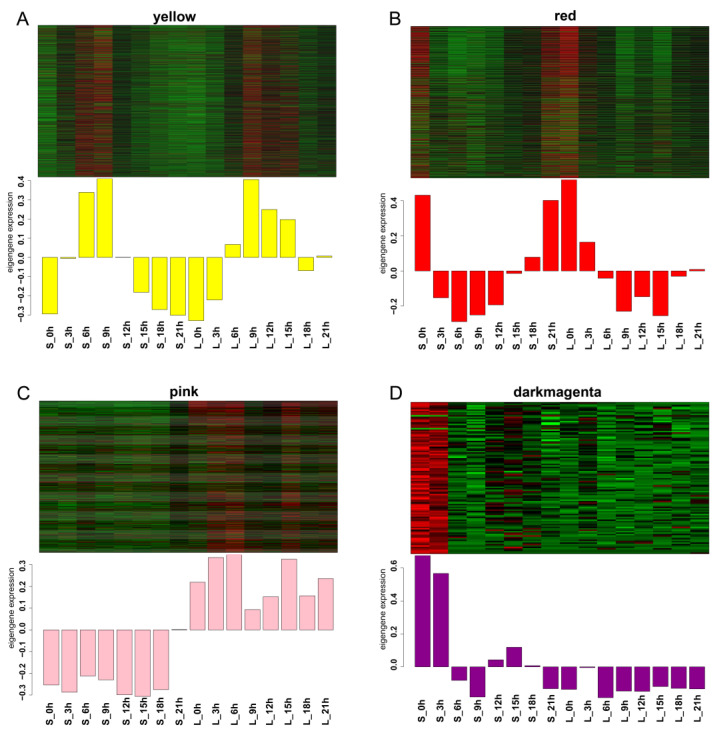
Heatmap of genes in the four modules. (**A**): Yellow module; (**B**): red module; (**C**): pink module; (**D**): dark-magenta module. S indicates short day; L indicates long day.

**Figure 10 ijms-25-01420-f010:**
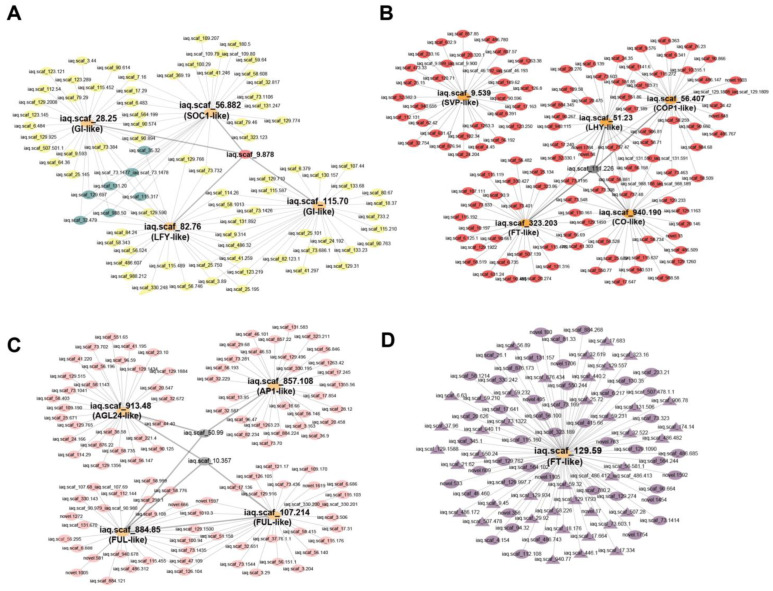
Co-expression network analysis of the flowering-associated genes in the four modules. (Ovals represent genes, and triangles represent transcription factors.) (**A**): Yellow module; (**B**): red module; (**C**): pink module; (**D**): dark magenta module. S specifies a short day; L indicates a long day.

**Table 1 ijms-25-01420-t001:** Hub gene transcription factor information in four modules.

Gene ID	Module	TF Family
iaq.scaf_90.574	yellow	BTB
iaq.scaf_605.27	yellow	Hist_deacetyl
iaq.scaf_20.597	yellow	Pribosyltran
iaq.scaf_90.957	yellow	RRM_1
iaq.scaf_32.479	yellow	2-Hacid_dh_C
iaq.scaf_9.878	yellow	G6PD_C
iaq.scaf_115.452	yellow	Pkinase
iaq.scaf_107.111	red	Pkinase_Tyr
iaq.scaf_507.139	red	DHBP_synthase
iaq.scaf_129.1922	red	Auxin_resp
iaq.scaf_330.427	red	Aminotran_1_2
iaq.scaf_90.203	red	FAD_binding_8
iaq.scaf_10.197	red	START
iaq.scaf_73.1195	red	Actin
iaq.scaf_47.109	pink	Actin
iaq.scaf_129.318	pink	ATP-synt_D
iaq.scaf_330.200_iaq.scaf_330.201	pink	JmjC
iaq.scaf_940.73	pink	HMG_box
iaq.scaf_41.108	pink	Aminotran_1_2
iaq.scaf_486.312	pink	3_5_exonuc
iaq.scaf_131.436	pink	Lipase_GDSL
iaq.scaf_20.568	pink	Kinesin
iaq.scaf_36.9	pink	FAD_binding_3
iaq.scaf_59.231	dark magenta	Cu_bind_like
iaq.scaf_59.232	dark magenta	Cu_bind_like
iaq.scaf_25.72	dark magenta	p450
iaq.scaf_17.334	dark magenta	p450
iaq.scaf_446.1	dark magenta	p450
iaq.scaf_6.63	dark magenta	p450
iaq.scaf_32.522	dark magenta	p450
iaq.scaf_129.934	dark magenta	Pkinase_Tyr
iaq.scaf_59.210	dark magenta	LRRNT_2
iaq.scaf_56.581.1	dark magenta	p450
iaq.scaf_29.92	dark magenta	Sugar_tr

## Data Availability

The original data presented in this study are publicly available. RNA-seq data were deposited at the NCBI under PRJNA1047901.

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
