# Peer review of "Evolution and Dynamic Transcriptome of Key Genes of Photoperiodic Flowering Pathway in Water Spinach (*Ipomoea aquatica*)"

_ijms, 2024, doi:10.3390/ijms25031420_

Round 1
Reviewer 1 Report
Comments and Suggestions for Authors
The manuscript describes the research aimed to identification of circadian rhytm-related genes in order to identify the photoperiod-mediated molecular regulation mechanism of flowering time in water spinach.
The aim of the work and the research concept is very good, but the methodological description and the description and presentation of the results raise many remarks and doubts (details can be found below).
Main remarks:
1. The methodology was described very briefly, without any details or even references to literature. Many significant details were missing, especially regarding the bioinformatics, phylogenetic and WGCNA analysis of the data.
2. The results chapter requires rewording, there are many inaccuracies and omissions - details are described below.
3. English must be improved – I recommend doing proofreading by native speaker.
Dataied remarks:
1. Title:
I don’t understand the meaning of “dynamic transcriptome of key genes” – I suggest doing improvement of the title.
2. In Abstract:
The abstract is not very informative; I see no point in the abstract relating the results obtained for water spins to Arabidopsis thaliana. The authors should pay more attention to the results presented in this publication.
In Line 20: please remove ....”including 48 samples”
Lines 20-21: Do the gene numbers refer to all identified DEGs, or are they only genes assigned to rhytm-related genes (via enrichment)? Under short and long day 8 time-points were analyzed – so for which time-point do these numbers refer to?
Lines 23-24: ..”conservatism of the reverse feedback pattern” – unclear
Line 25, line 27: in Molecular Phylogenetics gene duplication is used....please change “were amplified” into “were duplicated”
Lines 29-30: unclear statement
Line 31: compared to which species?
3. Introduction:
The entire chapter requires rewording, especially the beginning of the chapter (lines 39-78) – there are repetitions, lack of continuity of thought – below there are some examples.
Lines 53-54: Maybe this sentence should be moved to next paragraph (Lines 60-65)
Lines 74-78: Maybe this sentence should be incorporated with next paragraph (Lines 79-80)
Line 46: please delete one word growth
Line 51: Arabidopsis thaliana – please change into italic
Full gene names are missing i.e. if we are writing about a gene for the first time, there should be the full name of the protein encoded by the gene, and later an abbreviation can be used.
Lines 92-94: This sentence is not clear.
4. Materials and methods:
Line 146: water spinach was propagated by liquid culture – please give details (ingredients, etc.)
Line 154: young leaf samples – what do you mean? more detailed description
Chapter 2.2: methodology should be more detailed, please add: RNA extraction method, cDNA library construction (reagents), name of sequencing platform and mode RNA-seq (PE or SE reads?, length of reads), parameters of trimming process of reads, references for software (web page or article), data for reference genome used for mapping, parameters for mapping using Hisat2
The authors should provide in the methodology Bioproject, Biosample numbers for registered sequencing project in the SRA NCBI.
Line 161: what do you mean – “ploy-N”?
Line 162: The name and accession numbers of reference genome and version of annotation file should be provided
Line 166: provide reference for program
Lines 171-172: For enrichment, please provide p value. Lack of reference for ClusterProfiler program
Lines 176, 180: reference for programs
Chapter 2.6. Bioedit – which version? Which program was used for alignment i.e. Clustal or Muscle? Homologous sequences for which genes – please provide list of genes and homologs (Accession numbers). For construction of phylogenetic tree which substitution model was used?
Chapter 2.7. Lack of detailed methodology (reagents, qPCR conditions, list of selected genes and primers).
5. Results:
Lines 1194-195: a sentence unnecessarily repeated from the Material and Methods chapter
In the Supplementary materials I can’t find Table S1
Lines 203-205: Please move to the Material and Methods chapter
Lines 205-207: In my opinion all DEGs shouldn't be named as photoperiod-related genes because not all are truly related to photoperiod. This is especially true since the authors do not provide any information in the methodology or results as to how this number was calculated, especially since a total of 48 libraries, 24 for short and long day, were analyzed.
Figure 1: The results presented in the graphs refer to the Short-day vs. Long-day, but for which time-point? Propose to replace photoperiod-related genes with DEGs? The descriptions are illegible.
Chapter 3.3-3.4: Based on comparative analysis (MSA), the authors write that they identified gene copies and here I have questions and doubts:
- aren't these different transcription variants rather than gene copies (moreover, the authors once write about gene copies, in line 273 about subgroup members ??
- protein sequences were used for the phylogenetic analysis (Predicted i.e. XP_019159883.1) and sequence numbers that I have not been able to find in NCBI, so it is difficult to talk about homologs (paralogs and orthologs) here – hence the results described in the publication regarding the number of gene copies are, in my opinion, unreliable.
- In the Figure 6A I notice nucleotide sequences were used, but also described as predicted that’s why without methodology described in details I don’t understand different approaches used for preparation of Phylogenetic trees presented in Figure 3,5 and 6A.
Lines 270-272: Please move to Discussion section
Lines 278-280: “The homozygous iaq.scaf_9.675 locus was confirmed by Ilumina reads” – It’s not clear! Please give details about this analysis in methodology and results
For me it's not certain whether the sequence: https://www.ncbi.nlm.nih.gov/nuccore/XR_002055520.1/ can be confidently called ortologous sequences.
Figure 4: It was quoted only on line 299. On what basis was this data obtained? Gene abbreviations should be described with full gene names in the figure legend.
Chapter 3.6: The first sentence is not needed (the same the is in the methodology)
Figure 8: Graphs and descriptions are too small, they are not readable!
Figure 9 and Table 1: Description of the graph must be described in detail – what is it yellow, red etc. modules – for readers such a description is completely unclear.
Line 324: It should be error
Lines 325-350: The results are certainly very interesting, but the way they are described makes them completely unclear. Please edit this chapter.
Figure 10. Illegible
In summary: because of the comprehensive analysis of rich transcriptomic data I suggest that the manuscript could be considered to accept but after major changes and rewriting especially of the methodology and result parts.

English must be improved – I recommend doing proofreading by native speaker.
Author Response
The manuscript describes the research aimed to identification of circadian rhytm-related genes in order to identify the photoperiod-mediated molecular regulation mechanism of flowering time in water spinach.
The aim of the work and the research concept is very good, but the methodological description and the description and presentation of the results raise many remarks and doubts (details can be found below).
Main remarks:
- The methodology was described very briefly, without any details or even references to literature. Many significant details were missing, especially regarding the bioinformatics, phylogenetic and WGCNA analysis of the data.
Response back: Dear reviewer thank you for your remarks. We have briefly described material and method section and also provided the suitable reference for the mentioned section
- The results chapter requires rewording, there are many inaccuracies and omissions - details are described below.
Response back: We have tried to revise the manuscript. In addition. Further we have revised the manuscript English.
- English must be improved – I recommend doing proofreading by native speaker.
Response back: revised the manuscript English from horticulturist expert. Thank you for the manuscript improvement.
Dataied remarks:
- Title:
I don’t understand the meaning of “dynamic transcriptome of key genes” – I suggest doing improvement of the title.
Response back: We have revised the manuscript titled as per suggestion.
- In Abstract:
The abstract is not very informative; I see no point in the abstract relating the results obtained for water spins to Arabidopsis thaliana. The authors should pay more attention to the results presented in this publication.
Response back: L19-28:We have revised the abstract part and revised result section in abstract.
In Line 20: please remove ....”including 48 samples”
Response back: We have removed as per your direction.
Lines 20-21: Do the gene numbers refer to all identified DEGs, or are they only genes assigned to rhytm-related genes (via enrichment)? Under short and long day 8 time-points were analyzed – so for which time-point do these numbers refer to?
Response back: Response back: At 8 time points under long day conditions, all the different genes obtained by pairwise comparison were combined, and they were rhythm-related genes under long day conditions; At 8 time points under short day conditions, all the different genes obtained by pairwise comparison were combined, and they were rhythm-related genes under short day conditions; The number is the union of genes produced by pair-to-pair comparisons at eight time points.
Lines 23-24: ..”conservatism of the reverse feedback pattern” – unclear
Response back: CCA1/LHY positively regulates the expression of TOC1, and TOC1 also regulates the expression of CCA1/LHY
Line 25, line 27: in Molecular Phylogenetics gene duplication is used....please change “were amplified” into “were duplicated”
Response back: We have changed.
Lines 29-30: unclear statement
Response back: We have revised the statement.
Line 31: compared to which species?
Response back: I. nil, I. triloba, I. trifida
- Introduction:
The entire chapter requires rewording, especially the beginning of the chapter (lines 39-78) – there are repetitions, lack of continuity of thought – below there are some examples.
Response back: We have tried to revise manuscript, and as well as revised interdiction and below mentioned lines.
Lines 53-54: Maybe this sentence should be moved to next paragraph (Lines 60-65)
Response back: You are right, we have changed the sentence position.
Lines 74-78: Maybe this sentence should be incorporated with next paragraph (Lines 79-80)
Response back: You are right, we have changed the sentence position.
Line 46: please delete one word growth
Response back: We have deleted, thank you correction.
Line 51: Arabidopsis thaliana – please change into italic
Response back: We have changed, remove typos
Full gene names are missing i.e. if we are writing about a gene for the first time, there should be the full name of the protein encoded by the gene, and later an abbreviation can be used.
Response back: We have revised the whole manuscript.
Lines 92-94: This sentence is not clear.
Response back: We have revised the sentences.
- Materials and methods:
Line 146: water spinach was propagated by liquid culture – please give details (ingredients, etc.)
Response back: It was typo, correct term is hydroponic condition, experiment conducted under hydroponic environment, we have mentioned in the manuscript.
Line 154: young leaf samples – what do you mean? more detailed description
Response back: We have removed typos. Replaced with fresh leaf samples.
Chapter 2.2: methodology should be more detailed, please add: RNA extraction method, cDNA library construction (reagents), name of sequencing platform and mode RNA-seq (PE or SE reads?, length of reads), parameters of trimming process of reads, references for software (web page or article), data for reference genome used for mapping, parameters for mapping using Hisat2
Response back: We have explained the methodology section, detailed in brief.
The authors should provide in the methodology Bioproject, Biosample numbers for registered sequencing project in the SRA NCBI.
Response back: We have mentioned in the manuscript also provided here “PRJNA1047901”
Line 161: what do you mean – “ploy-N”?
Response back: Remove reads containing N (N indicates that base information cannot be determined).
Line 162: The name and accession numbers of reference genome and version of annotation file should be provided
Response back: The reference genome was deposited at the BIGD under PRJCA002216.
Line 166: provide reference for program
Response back: We have provided the programmed referenced.
Lines 171-172: For enrichment, please provide p value. Lack of reference for Cluster Profiler program
Response back: We have provided the reference.
Lines 176, 180: reference for programs
Response back: We have provided the reference.
Chapter 2.6. Bioedit – which version? Which program was used for alignment i.e. Clustal or Muscle? Homologous sequences for which genes – please provide list of genes and homologs (Accession numbers). For construction of phylogenetic tree which substitution model was used?
Response back: Bioedit – which version(v7.2.5.0), for construction of phylogenetic tree used
neighbor-joining method.
Chapter 2.7. Lack of detailed methodology (reagents, qPCR conditions, list of selected genes and primers).
Response back: We have provided the detailed methodology. And also provided the list of primer in supplementary file.
- Results:
Lines 194-195: a sentence unnecessarily repeated from the Material and Methods chapter
Response back: We have removed the sentence from upapproperiate places.
In the Supplementary materials I can’t find Table S1
Response back: We have provided the table S1.
Lines 203-205: Please move to the Material and Methods chapter
Response back: We have moved this line to
Lines 205-207: In my opinion all DEGs shouldn't be named as photoperiod-related genes because not all are truly related to photoperiod. This is especially true since the authors do not provide any information in the methodology or results as to how this number was calculated, especially since a total of 48 libraries, 24 for short and long day, were analyzed.
Response back: We divided long-day samples (24 samples) and short-day samples (24 samples) into two groups, and analyzed the differences between the groups. The genes obtained were named photoperiod related genes.
Figure 1: The results presented in the graphs refer to the Short-day vs. Long-day, but for which time-point? Propose to replace photoperiod-related genes with DEGs? The descriptions are illegible.
Response back: We divided the long-day samples (24 samples) and the short-day samples (24 samples) into two groups. The figure represents the long-day and short-day samples
Chapter 3.3-3.4: Based on comparative analysis (MSA), the authors write that they identified gene copies and here I have questions and doubts:
- aren't these different transcription variants rather than gene copies (moreover, the authors once write about gene copies, in line 273 about subgroup members ??
Response back: The gene copy here refers to one gene in Arabidopsis, while two homologous genes appear in water spinach.
- protein sequences were used for the phylogenetic analysis (Predicted i.e. XP_019159883.1) and sequence numbers that I have not been able to find in NCBI, so it is difficult to talk about homologs (paralogs and orthologs) here – hence the results described in the publication regarding the number of gene copies are, in my opinion, unreliable.
Response back: The XP_019159883.1 can be found in NCBI.
- In the Figure 6A I notice nucleotide sequences were used, but also described as predicted that’s why without methodology described in details I don’t understand different approaches used for preparation of Phylogenetic trees presented in Figure 3,5 and 6A.
Response back: Using nucleotide sequences to analyze why iaq.scaf_9.675 are shorter
Phylogenetic trees are constructed using amino acid sequences.
Lines 270-272: Please move to Discussion section
Response back: We have moved to discussion section, thank you for correction.
Lines 278-280: “The homozygous iaq.scaf_9.675 locus was confirmed by Ilumina reads” – It’s not clear! Please give details about this analysis in methodology and results
Response back: We have removed the sentence upon other reviewer suggestion.
For me it's not certain whether the sequence: https://www.ncbi.nlm.nih.gov/nuccore/XR_002055520.1/ can be confidently called ortologous sequences.
Response back: This is a homologous gene obtained by sequence alignment
Figure 4: It was quoted only on line 299. On what basis was this data obtained? Gene abbreviations should be described with full gene names in the figure legend.
Response back: This is a diagram of flowering regulation in Arabidopsis Thaliana, showing the number of genes in different species
Chapter 3.6: The first sentence is not needed (the same the is in the methodology)
Response back: We have removed the sentence
Figure 8: Graphs and descriptions are too small, they are not readable!
Response back: We have revised it.
Figure 9 and Table 1: Description of the graph must be described in detail – what is it yellow, red etc. modules – for readers such a description is completely unclear.
Response back: We have revised it.
Line 324: It should be error
Response back: We have removed.
Lines 325-350: The results are certainly very interesting, but the way they are described makes them completely unclear. Please edit this chapter.
Response back: We have tried to revise the manuscript as well as revised the results.
Figure 10. Illegible
Response back: We have revised the figure.
In summary: because of the comprehensive analysis of rich transcriptomic data I suggest that the manuscript could be considered to accept but after major changes and rewriting especially of the methodology and result parts.
Response back: Thank you for your kind remarks.

Reviewer 2 Report
Comments and Suggestions for Authors
In the manuscript “Evolution and dynamic transcriptome of key genes of photoperiodic flowering pathway in water spinach (Ipomoea aquatica)”, the authors studied genes that were differentially expressed at different time points during two photoperiod treatment (short and long days) in water spinach to provide better understanding of the photoperiod-mediated molecular regulation mechanism of flowering time.
Experiments and data analysis were well done, and the study design was appropriate to answer their biological questions, but in the manuscript text some explanations are needed. Further clarification of how certain procedures were performed is necessary in the materials and methods section for the readers to validate their results. For example, authors are not providing important information on primers used to validate gene expression by qPCR, among other details). Furthermore, the cohesion and coherence of ideas in the results section needs to be improved so the reader understands the author analysis and conclusions. The points to be clarified and improved can be divided into major and minor issues as described below:
1. Title: Ipomoea aquatica in title is a scientific name and the convention for scientific name is that it needs to be italicized.
2. Introduction (page 2, line 51): Arabidopsis thaliana is a scientific name and the convention for scientific name is that it needs to be italicized.
3. Materials and methods (page 3, line 146): How was water spinach propagated by liquid culture? At what age were the plants treated? Please provide details.
4. Materials and methods (page 4, line 158-162): How was the total RNA extracted? What kind of library was constructed? And what Illumina instrument was used for sequencing? Please provide details. Also, software used for analysis (for example: Hisat2 v2.0.5) needs to have a citation. Which software was used to “removing reads containing adapter, reads containing ploy-N and low-quality reads from raw data”? What is ploy-N? Not sure what that term means, is this a typo?
5. Materials and methods (page 4, line 165): Citation is needed for the package “DESeq2 R package (1.20.0)”.
6. Materials and methods (page 4, line 172): Citation is needed for the package “cluster Profiler R package”.
7. Materials and methos (page 4, lines 176-179): Citations needed for the package “the WGCNA package in R” and the software “Cytoscape 179 (v.3.8.1)“.
8. Materials and methods (page 4, line 182-183): Citations are needed for “Bioedit software” and “MEGA 8.0”.
9. Materials and methods (page 4, line 186-191): What genes were selected when author indicate “10 flowering-related genes for RT‐qPCR validation” Not clear. Which primers were used? Please include a table with gene, primer (F and R) name, primer (F and R) sequences, and corresponding melting temperature used during PCR. Please provide details on how the total RNA was retrotranscribed to cDNA and how the qPCR was performed? (SYBR Green kit? FAM kit? Instrument? Temperature conditions? Melting curve analysis? How many biological replicates per sample? among other important details for qPCR). The materials and methods section must be very specific so other people wanting to validate the results can do it exactly in the same way authors did.
10. Results (page 4, line 199): Ipomoea aquatica is a scientific name and the convention for scientific name is that it needs to be italicized. What version of the reference genome was used for alignment? Also, where is the reference genome found (database)? Authors did not mention these details in the materials and methods section. Please specify it and provide more details in the materials and methods section.
11. Results (page 4, line 201): Table S1 is missing from the materials provided by the authors, so I have no way to check this table. Please, for future reviewers, provide the table, otherwise is like the table does not exist.
12. Results (page 5, Figure 1): Font needs to be bigger in all three panels, so it can be readable in a printed version of the manuscript.
13. Results (page 7, lines 253-256): Not clear if the paragraph refers to the results obtained in water spinach or Arabidopsis. Please clarify. Also, this paragraph need citations? It seems like is part of a discussion and not the results section. Please remove from results if it belongs to discussion. Not sure what the purpose of the paragraph is. The purpose is not clear.
14. Results (page 7, Figure 3): Does S represents short days and L represent Long days? Please clarify in the figure description. And add it to all figures that have the S and L presented as well.
15. Results (page 8, Figure 4): Not clear where Figure 4 is referenced to in the results? What paragraph of the results described figure 4?
16. Results (page 8, lines 270-272): Not clear if the paragraph refers to the results obtained in water spinach or Arabidopsis. Please clarify. Also, this paragraph need citations? It seems like is part of a discussion and not the results section. Please remove from results if it belongs to discussion. Not sure what the purpose of the paragraph is. The purpose is not clear.
17. Results (page 8, line 299) Figure 4 was presented 2 figures ago and cited until now? Not clear what the purpose of Figure 4 really is. Please cite figure 4 in the results where it corresponds to cite it. It is confusing citing a figure later in the text when it was presented long ago.
18. Results (page 10, Figure 7): Font needs to be bigger in panel D, so it can be readable in a printed version of the manuscript.
19. Results (page 10, no line number indicated here, please add the lines number to facilitate reviewer work): What do the authors mean by “meanwhile, SVP, AGL24, AP1, and FKF1 have no homologs in C. australis (Figure S1).” Where does this new species C. australis comes from? Results should refer to water spinach, not a new species. This is very confusing, please fix this.
20. Results (referring to all supplementary figures in the word document): No supplementary figure has a description under the figure name. Please add a description so readers can know why the figure is important to see.
21. Results (page 10, no line number here, please add the lines number to facilitate reviewer work): Is FKFI the same as FPK1? Not clear where the results of this paragraph are found. All the paragraphs found in page 10 need to refer to where the results are found. Not just at the very end of the section in page 11. Cohesion and coherence of the section “3.4. Analysis of photoperiod-mediated flowering-related genes” needs to be improved, right now, as it is, it is not clear at all what the purpose of the author is here.
22. Results (page 11, section 3.5): Authors indicate here that “eight flowering-related genes for qRT-PCR analysis to verify the accuracy of RNA-seq results” but the materials and methods indicate ten (Materials and methods, page 4, line 186: “10 flowering-related genes for RT‐qPCR validation”). What was it, ten or eight? Not clear.
23. Results (page 11, section 3.6): Authors indicate “WGCNA analysis constructs a hierarchical clustering tree based on the correlation between the expression levels of genes and divided divides the different modules, the genes in the same module may have similar biological functions.” Not clear what divided divides means. Is this a typo? Please fix to improve clarity of ideas.
24. Results (page 12, Figure 8): Font needs to be bigger in all three panels, so it can be readable in a printed version of the manuscript. What about correlations with light time? Not sure why correlations with long and short days are presented together in the same column in panel B. Shouldn’t this be presented separately? What is the number in parenthesis? What does the colors in the scale are? Expression levels? Correlation level? Please clarify.
25. Results (page 15, line 324): Authors indicate “A: yellow module; B: red module; C: pink module; D: darkmagenta module” What is this? Does this correspond to a supplementary figure description? Please see my previous comment on all supplementary figures, they all need a description with their names.
26. Results (page 16, Figure 10): Font needs to be bigger in all the four panels, so it can be readable in a printed version of the manuscript. Please indicate what the difference between circles and triangles is in panel D. The quality needs to be improved, even zooming in in the figure, the letters disappear.
27. Discussion (page 17, line 391): Arabidopsis thaliana is a scientific name and the convention for scientific name is that it needs to be italicized.
Comments on the Quality of English LanguageTake home messages from Section 3.5 and 3.6 (results) are not really clear. the cohesion and coherence of ideas here needs to be improved and the authors purpose on these sections needs to be presented in a way that is clear what they did and why they did it, otherwise these sections are really difficult to understand and follow as a reader.
Author Response
In the manuscript “Evolution and dynamic transcriptome of key genes of photoperiodic flowering pathway in water spinach (Ipomoea aquatica)”, the authors studied genes that were differentially expressed at different time points during two photoperiod treatment (short and long days) in water spinach to provide better understanding of the photoperiod-mediated molecular regulation mechanism of flowering time.
Experiments and data analysis were well done, and the study design was appropriate to answer their biological questions, but in the manuscript text some explanations are needed. Further clarification of how certain procedures were performed is necessary in the materials and methods section for the readers to validate their results. For example, authors are not providing important information on primers used to validate gene expression by qPCR, among other details). Furthermore, the cohesion and coherence of ideas in the results section needs to be improved so the reader understands the author analysis and conclusions. The points to be clarified and improved can be divided into major and minor issues as described below:
Response back: Dear reviewer, thank you for your time and important suggestion, your valuable suggestion; improved the manuscript quality. Thank you.
- Title: Ipomoea aquatica in title is a scientific name and the convention for scientific name is that it needs to be italicized.
Response back: We have removed the scientific name from the title, upon suggestion of other reviewers.
- Introduction (page 2, line 51): Arabidopsis thaliana is a scientific name and the convention for scientific name is that it needs to be italicized.
Response back: We have changed into italicized.
- Materials and methods (page 3, line 146): How was water spinach propagated by liquid culture? At what age were the plants treated? Please provide details.
Response back: It was typo, we have corrected, propagated by hydroponic methods, experiment conducted under hydroponic conditions.
- Materials and methods (page 4, line 158-162): How was the total RNA extracted? What kind of library was constructed? And what Illumina instrument was used for sequencing? Please provide details. Also, software used for analysis (for example: Hisat2 v2.0.5) needs to have a citation. Which software was used to “removing reads containing adapter, reads containing ploy-N and low-quality reads from raw data”? What is ploy-N? Not sure what that term means, is this a typo?
Response back: We have mentioned all detailed in material and method section. We have provided the complete procedure of RNA extracted.
- Materials and methods (page 4, line 165): Citation is needed for the package “DESeq2 R package (1.20.0)”.
Response back: We have provided the suitable reference
- Materials and methods (page 4, line 172): Citation is needed for the package “cluster Profiler R package”.
Response back: We have provided the suitable reference,
- Materials and methods (page 4, lines 176-179): Citations needed for the package “the WGCNA package in R” and the software “Cytoscape 179 (v.3.8.1)“.
Response back: We have provided the suitable reference
- Materials and methods (page 4, line 182-183): Citations are needed for “Bioedit software” and “MEGA 8.0”.
Response back: We have provided the suitable reference
- Materials and methods (page 4, line 186-191): What genes were selected when author indicate “10 flowering-related genes for RT‐qPCR validation” Not clear. Which primers were used? Please include a table with gene, primer (F and R) name, primer (F and R) sequences, and corresponding melting temperature used during PCR. Please provide details on how the total RNA was retrotranscribed to cDNA and how the qPCR was performed? (SYBR Green kit? FAM kit? Instrument? Temperature conditions? Melting curve analysis? How many biological replicates per sample? among other important details for qPCR). The materials and methods section must be very specific so other people wanting to validate the results can do it exactly in the same way authors did.
Response back: We have provided the all details in material and methods section. Thank you for intension.
- Results (page 4, line 199): Ipomoea aquatica is a scientific name and the convention for scientific name is that it needs to be italicized. What version of the reference genome was used for alignment? Also, where is the reference genome found (database)? Authors did not mention these details in the materials and methods section. Please specify it and provide more details in the materials and methods section.
Response back: We have changed italics,The reference genome was deposited at the BIGD under PRJCA002216.
- Results (page 4, line 201): Table S1 is missing from the materials provided by the authors, so I have no way to check this table. Please, for future reviewers, provide the table, otherwise is like the table does not exist.
Response back: We have provided the table S1. Sorry for inconvenience.
- Results (page 5, Figure 1): Font needs to be bigger in all three panels, so it can be readable in a printed version of the manuscript.
Response back: We have revised the figures 1.
- Results (page 7, lines 253-256): Not clear if the paragraph refers to the results obtained in water spinach or Arabidopsis. Please clarify. Also, this paragraph need citations? It seems like is part of a discussion and not the results section. Please remove from results if it belongs to discussion. Not sure what the purpose of the paragraph is. The purpose is not clear.
Response back: We have revised the figure, also revised the results section and also moved to discussion section.
- Results (page 7, Figure 3): Does S represents short days and L represent Long days? Please clarify in the figure description. And add it to all figures that have the S and L presented as well.
Response back: We have indicated in all the figures.
- Results (page 8, Figure 4): Not clear where Figure 4 is referenced to in the results? What paragraph of the results described figure 4?
Response back: We have revised the figure resolution, and described the figures in result section.
- Results (page 8, lines 270-272): Not clear if the paragraph refers to the results obtained in water spinach or Arabidopsis. Please clarify. Also, this paragraph need citations? It seems like is part of a discussion and not the results section. Please remove from results if it belongs to discussion. Not sure what the purpose of the paragraph is. The purpose is not clear.
Response back: We have moved the paragraph relavent materials to discussion section.
- Results (page 8, line 299) Figure 4 was presented 2 figures ago and cited until now? Not clear what the purpose of Figure 4 really is. Please cite figure 4 in the results where it corresponds to cite it. It is confusing citing a figure later in the text when it was presented long ago.
Response back: We have revised the figures and updated the figure sequences. Also cited the figure 4.
- Results (page 10, Figure 7): Font needs to be bigger in panel D, so it can be readable in a printed version of the manuscript.
Response back: We have revised the figure and increased figure resolution.
- Results (page 10, no line number indicated here, please add the lines number to facilitate reviewer work): What do the authors mean by “meanwhile, SVP, AGL24, AP1, and FKF1 have no homologs in C. australis (Figure S1).” Where does this new species C. australis comes from? Results should refer to water spinach, not a new species. This is very confusing, please fix this.
Response back: The sentence was deleted.
- Results (referring to all supplementary figures in the word document): No supplementary figure has a description under the figure name. Please add a description so readers can know why the figure is important to see.
Response back: We have described all figure in the results also added all supplementary files.
- Results (page 10, no line number here, please add the lines number to facilitate reviewer work): Is FKFI the same as FPK1? Not clear where the results of this paragraph are found. All the paragraphs found in page 10 need to refer to where the results are found. Not just at the very end of the section in page 11. Cohesion and coherence of the section “3.4. Analysis of photoperiod-mediated flowering-related genes” needs to be improved, right now, as it is, it is not clear at all what the purpose of the author is here.
Response back: It was typo, I have corrected the gene name, we have revised the result section.
- Results (page 11, section 3.5): Authors indicate here that “eight flowering-related genes for qRT-PCR analysis to verify the accuracy of RNA-seq results” but the materials and methods indicate ten (Materials and methods, page 4, line 186: “10 flowering-related genes for RT‐qPCR validation”). What was it, ten or eight? Not clear.
Response back: It was typo, 10 flowering genes were selected for RNA-seq analysis. We have also provide supplementary file.
- Results (page 11, section 3.6): Authors indicate “WGCNA analysis constructs a hierarchical clustering tree based on the correlation between the expression levels of genes and divided divides the different modules, the genes in the same module may have similar biological functions.” Not clear what divided divides means. Is this a typo? Please fix to improve clarity of ideas.
Response back: It was typo, we have corrected the sentence and lines, thank you for intension.
- Results (page 12, Figure 8): Font needs to be bigger in all three panels, so it can be readable in a printed version of the manuscript. What about correlations with light time? Not sure why correlations with long and short days are presented together in the same column in panel B. Shouldn’t this be presented separately? What is the number in parenthesis? What does the colors in the scale are? Expression levels? Correlation level? Please clarify.
Response back: The colors represent different modules, and the numbers represent correlations.
- Results (page 15, line 324): Authors indicate “A: yellow module; B: red module; C: pink module; D: darkmagenta module” What is this? Does this correspond to a supplementary figure description? Please see my previous comment on all supplementary figures, they all need a description with their names.
Response back: We have removed the unnecessary lines, it was typo, we have added the description in the supplementary files.
- Results (page 16, Figure 10): Font needs to be bigger in all the four panels, so it can be readable in a printed version of the manuscript. Please indicate what the difference between circles and triangles is in panel D. The quality needs to be improved, even zooming in in the figure, the letters disappear.
Response back: We have improved the figure quality and figure resolution, easy for readers.
- Discussion (page 17, line 391): Arabidopsis thaliana is a scientific name and the convention for scientific name is that it needs to be italicized.
Response back: We have changed into italicized

Reviewer 3 Report
Comments and Suggestions for Authors
In my opinion, the present research on the identification of genes related to flowering and their association with photoperiod in water spinach is interesting. Although the research seems to be well done and to provide conclusive results, on a methodological level it should be better explained and the terms included in the results and not mentioned in the Material and Methods should be completed. Crop details and sampling details should be included. At the background level, it seems acceptable, although it could be improved, although there is an imbalance between the introduction and the discussion, so that the discussion needs to be expanded and the introduction needs to be refined. At the results level, significant differences should be included in the comparative graphs, the legends should be improved to make the graphs immediately understandable, and the format and size of the figures should be adjusted. The discussion should be improved. The rest of the comments are included in the attached document.

In my opinion, the English language should be revised so that it is written in a more linear way.
Author Response
In my opinion, the present research on the identification of genes related to flowering and their association with photoperiod in water spinach is interesting. Although the research seems to be well done and to provide conclusive results, on a methodological level it should be better explained and the terms included in the results and not mentioned in the Material and Methods should be completed. Crop details and sampling details should be included. At the background level, it seems acceptable, although it could be improved, although there is an imbalance between the introduction and the discussion, so that the discussion needs to be expanded and the introduction needs to be refined. At the results level, significant differences should be included in the comparative graphs, the legends should be improved to make the graphs immediately understandable, and the format and size of the figures should be adjusted. The discussion should be improved. The rest of the comments are included in the attached document.
Response: Dear reviewer thank you for your valuable remarks and kind suggestion. As per your direction, we have revised the introduction and also tried to improved discussion section. We have also revised the manuscript figures, and as well as revised figure legend and figure resolution.
In addition, we have also revise the all suggested comments as we have received in the attachment.
At the end, I am also thankful to editor and reviewer for this valuable suggestion and improvement.
Line 41 42: your suggestion in pdf “I think both phrases could be put together.”
Response: Thank you for attention, if we put together sentence will be too long, we are considering as two sentence. Thank you.

Round 2
Reviewer 3 Report
Comments and Suggestions for Authors
In my opinion, after a second review, I have missed a detailed response to my comments. However, I believe that significant improvements have been made. Indicate that "Ipomoea genus" (line 39) is a redundancy. The conditions of the "hydroponic method" should be explained. In lines 201-203 the terms of the formulas should be explained or eliminated once the reference has been indicated. Terms such as "Q30" or "hub genes" should be clearly explained in Materials and Methods. Consideration should be given to improving some of the legends and including the significant differences in the gene expression of the figures shown. Although the discussion is basically the same, and the introduction is a bit disproportionate, I consider that at the discretion of the editor, and after outlining the indicated details, it can be published.
Comments on the Quality of English LanguageIn my opinion, although in a previous view I indicates moderate revision, I consider that with only minor changes could be acceptable.